# Two Features of the GINAR(1) Process and Their Impact on the Run-Length Performance of Geometric Control Charts

**DOI:** 10.3390/e25030444

**Published:** 2023-03-02

**Authors:** Manuel Cabral Morais

**Affiliations:** Department of Mathematics & CEMAT (Center for Computational and Stochastic Mathematics), Instituto Superior Técnico, Universidade de Lisboa, 1049-001 Lisbon, Portugal; maj@math.ist.utl.pt

**Keywords:** discrete-time Markov chain, TP_2_ transition probability matrix, Kalmykov order, statistical process control, run length, 60J10, 60E15, 62P30

## Abstract

The geometric first-order integer-valued autoregressive process (GINAR(1)) can be particularly useful to model relevant discrete-valued time series, namely in statistical process control. We resort to stochastic ordering to prove that the GINAR(1) process is a discrete-time Markov chain governed by a totally positive order 2 $\left({TP}_{2}\right)$ transition matrix.Stochastic ordering is also used to compare transition matrices referring to pairs of GINAR(1) processes with different values of the marginal mean. We assess and illustrate the implications of these two stochastic ordering results, namely on the properties of the run length of geometric charts for monitoring GINAR(1) counts.

## 1. Introduction

The INAR(1) and GINAR(1) processes were originally proposed by McKenzie [1,2]; the latter model was soon after discussed in more detail by Alzaid and Al-Osh [3]. They rely on the binomial thinning operation due to Steutel and van Harn [4] which is defined below.

**Definition** **1.** 
*Let X be a non-negative integer-valued r.v. with range $N_{0}=\left\{0,1,\cdots \right\}$ and ρ a scalar in (0,1). Then the binomial thinning operation on X results in the r.v.*

(1)
$$\rho \circ X=\sum_{t=1}^{X}Y_{t},$$

*where ∘ represents the binomial thinning operator; $\left\{Y_{t}:t\in N\right\}$ is a sequence of i.i.d. Bernoulli r.v. with parameter ρ; $\left\{Y_{t}:t\in N\right\}$ is independent of X.*

*We usually refer to ρ∘X as the r.v. that arises from X by binomial thinning. Furthermore, we define 0∘X=0 and 1∘X=X.*


Now that we have defined the binomial thinning operation, a sort of scalar multiplication counterpart in the integer-valued setting, the reader is reminded of the definition of McKenzie’s GINAR(1) process and its main properties.

**Definition** **2.** 
*Let ρ,p∈(0,1). Then $\left\{X_{t}:t\in N_{0}\right\}$ is said to be a GINAR(1) process if $X_{t}$ is written in the form*

(2)
$$X_{t}=\rho \circ X_{t-1}+B_{t}\times G_{t},$$

*where $\left\{B_{t}:i\in N\right\}$ and $\left\{G_{t}:i\in N\right\}$ are independent sequences of i.i.d. Bernoulli r.v. with parameter (1−ρ) and of i.i.d. geometric r.v. with parameter p, respectively; the sequence of innovations $\left\{\varepsilon_{t}=B_{t}\times G_{t}:t\in t\in N\right\}$ and $\left\{X_{0},\cdots ,X_{t-2},X_{t-1}\right\}$ are independent; all thinning operations are performed independently of each other and of $\left\{\varepsilon_{t}:t\in N\right\}$; and all the thinning operations at time t are independent of $\left\{X_{0},\cdots ,X_{t-2},X_{t-1}\right\}$.*


According to McKenzie [2] and Alzaid and Al-Osh [3], if $X_{0}∼geometric\left(p\right)$ then $\left\{X_{t}:t\in N_{0}\right\}$ is a stationary AR(1) process with geometric(p) marginal distribution.

McKenzie [2] also adds that $\left\{X_{t}:t\in N_{0}\right\}$ is a DTMC with TPM, $P\left(p,\rho \right)={\left[p_{ij}\left(p,\rho \right)\right]}_{i,j\in N_{0}}$$={\left[P\left(X_{t}=j|X_{t-1}=i\right)\right]}_{i,j\in N_{0}}$, where
(3)pij(p,ρ)=∑m=0min{i,j}imρm(1−ρ)i−m×(1−ρ)(1−p)j−mp+ijρj(1−ρ)i−j×ρ×IN0(i−j),i,j∈N0,
where $I_{N_{0}}$ represents the indicator function of the set of non-negative integers. These entries can be obtained by taking advantage of a few facts: $(\rho \circ X_{t-1}|X_{t-1}=0)=0$ with probability 1; $\left(\rho \circ X_{t-1}|X_{t-1}=i\right)∼binomial\left(i,\rho \right)$, for i∈N; the p.f. of the innovations, $\varepsilon_{t}=B_{t}\times G_{t}$, is equal to
(4)$$P\left(\varepsilon_{t}=j\right)=\left\{\begin{array}{cc} P\left(B_{t}=0\;or\;G_{t}=0\right)=p\;\left(1-\rho \right)+\rho , & j=0\\ P\left(B_{t}=1,G_{t}=j\right)=\left(1-\rho \right)\;{\left(1-p\right)}^{j}p, & j\in N. \end{array}\right.$$

The autocorrelation function of the GINAR(1) process is equal to
(5)$$corr\left(X_{t},X_{t+k}\right)=\rho^{k},\;k,t\in N_{0}.$$

We ought to point out that the GINAR(1) process is a particular case of the generalized geometric INAR(1) or GGINAR(1) process, introduced by (Al-Osh and Aly [5], Section 3). Moreover, autocorrelated geometric counts can also be modeled by the new geometric INAR(1) or NGINAR(1) process, proposed by Ristić et al. [6] and relying on the negative binomial thinning operator. Finally, the NGINAR(1) process is a special instance of the ZMGINAR(1) process, the zero-modified geometric first-order integer-valued autoregressive, introduced and thoroughly described by Barreto-Souza [7].

The remainder of the paper is organized as follows. In Section 2, we shall prove that P has two important features stated in the two following theorems.

**Theorem** **1.** 
*The TPM P(p,ρ) of a GINAR(1) process is totally positive of order 2,*

(6)
$$P\left(p,\rho \right)\in {TP}_{2},$$

*i.e., all the 2×2 minors of the P(p,ρ) are non-negative.*


**Theorem** **2.** 
*Let: $\left\{X_{t}\left(p,\rho \right):t\in N_{0}\right\}$ and $\left\{X_{t}\left(p^{\prime },\rho \right):t\in N_{0}\right\}$ be two independent GINAR(1) processes, with parameters (p,ρ) and $\left(p^{\prime },\rho \right)$; $P\left(p,\rho \right)={\left[p_{ij}\left(p,\rho \right)\right]}_{i,j\in \{0,1,\cdots ,n\}}$ and $P\left(p^{\prime },\rho \right)={\left[p_{ij}\left(p^{\prime },\rho \right)\right]}_{i,j\in \{0,1,\cdots ,n\}}$ be their corresponding TPM. Then $P(p^{\prime },\rho )$ is stochastically smaller than P(p,ρ) in the usual (or in the Kalmykov order) sense,*

(7)
$$P\left(p^{\prime },\rho \right)\leq_{st}P\left(p,\rho \right),$$

*if $\;0\leq \sqrt{\rho }/\left(\sqrt{\rho }+1\right)<p\leq p^{\prime }<1$, that is,*

$$\sum_{j=l}^{n}p_{i\;j}\left(p,\rho \right)\leq \sum_{j=l}^{n}p_{m\;j}\left(p^{\prime },\rho \right),\;i,l,m\in \left\{0,1,\cdots ,n\right\},\;i\leq m,$$

*in case $1/\sqrt{\rho }>E\left[X_{t}\left(p,\rho \right)\right]\geq E\left[X_{t}\left(p^{\prime },\rho \right)\right]>0$.*


In Section 3, we discuss and illustrate the impact of (Equation 6) and (Equation 7) on the run length of an upper one-sided geometric chart for monitoring GINAR(1) processes. In Section 4, we sum up our findings and briefly refer to related and future work.

## 2. Proving the Two Features of the GINAR(1) Process

Demonstrating that the 2×2 minors of the TPM of a GINAR(1) process are all non-negative is not simple, due to the aspect of the transition probabilities defined in (Equation 3). However, by adopting the reasoning of (Morais and Pacheco [8] Section 2) and resorting to some auxiliary definitions and lemmas in Section A.1, we can prove (Equation 6).

**Proof**** ****of**** ****Theorem**** ****1.** Note that
$$\left(X_{t+1}\left(p,\rho \right)|X_{t}\left(p,\rho \right)=i\right)=\rho \circ X_{t-1}\left(p,\rho \right)+\varepsilon_{t}\overset{st}{=}B\left(i,\rho \right)+BG\left(p,\rho \right),$$
where: $B\left(0,\rho \right)\overset{st}{=}0$; B(i,ρ)∼binomial(i,ρ), i∈N; BG(p,ρ) a r.v. with p.f. given by (Equation 4); B(i,ρ) and BG(p,ρ) are two independent r.v.In accordance to Lemmas A1 and A3, B(i) stochastically increases with *i* in the likelihood ratio sense and $B\left(i,\rho \right),BG\left(p,\rho \right)\in {PF}_{2}$. Hence, we can invoke the closure of the stochastic order $\leq_{lr}$ (see Definition A1) under the sum of independent ${PF}_{2}$ r.v. (see (Shaked and Shanthikumar [9] p. 46, Theorem 1.C.9) or Karlin and Proschan [10]) to conclude that
$$\begin{array}{ccc} B(i,\rho )+BG(p,\rho ) & \leq_{lr} & B(i+1,\rho )+BG(p,\rho )\\ \left(X_{t+1}\left(p,\rho \right)|X_{t}\left(p,\rho \right)=i\right) & \leq_{lr} & \left(X_{t+1}\left(p,\rho \right)|X_{t}\left(p,\rho \right)=i+1\right), \end{array}$$
for $i\in N_{0}$, i.e., $P\in {TP}_{2}$ or P is a stochastically monotone TPM in the likelihood ratio sense $\left(P\in M_{lr}\right)$, according to Definition A2. □

The next proof refers to a stochastic ordering between the TPM that govern two DTMC with the same state space, thus associated with what Kulkarni [11] (pp. 148–149) terms the Kalmykov-dominance or Kalmykov order(see Kalmykov [12] Theorem 2).

**Proof**** ****of**** ****Theorem**** ****2.** Result (Equation 7) can be shown to hold by successively capitalizing on: Lemmas A1 and A2; the closure of $\leq_{lr}$ under the sum of independent ${PF}_{2}$ r.v.; $P\left(p^{\prime },\rho \right)\in {TP}_{2}$; and $X\leq_{lr}Y$ implies that the r.v. *X* is stochastically smaller than the r.v. *Y* in the usual sense, in short $X\leq_{st}Y$ (see Shaked and Shanthikumar [9] p. 42, Theorem 1.C.1). Then, for i,m∈{0,1,⋯,n}, i≤m, and $0\leq \sqrt{\rho }/\left(\sqrt{\rho }+1\right)<p\leq p^{\prime }<1$:
$$\begin{array}{ccc} \left(X_{t+1}\left(p^{\prime },\rho \right)|X_{t}\left(p^{\prime },\rho \right)=i\right) & \overset{st}{=} & B\left(i,\rho \right)+BG\left(p^{\prime },\rho \right)\\ & \leq_{lr} & B(i+1,\rho )+BG(p,\rho )\\ & \overset{st}{=} & \left(X_{t+1}\left(p,\rho \right)|X_{t}\left(p,\rho \right)=i\right)\\ & \leq_{lr} & \left(X_{t+1}\left(p,\rho \right)|X_{t}\left(p,\rho \right)=m\right)\\ \left(X_{t+1}\left(p^{\prime },\rho \right)|X_{t}\left(p^{\prime },\rho \right)=i\right) & \leq_{st} & \left(X_{t+1}\left(p,\rho \right)|X_{t}\left(p,\rho \right)=m\right)\\ \sum_{j=l}^{n}p_{i\;j}\left(p^{\prime },\rho \right) & \leq & \sum_{j=l}^{n}p_{m\;j}\left(p,\rho \right),\;l\in \left\{0,1,\cdots ,n\right\}, \end{array}$$
i.e., $P\left(p^{\prime },\rho \right)\leq_{st}P\left(p,\rho \right)$ if $1/\sqrt{\rho }>1/p-1=E\left[X_{t}\left(p,\rho \right)\right]\geq E\left[X_{t}\left(p^{\prime },\rho \right)\right]=1/p^{\prime }-1>0$. □

## 3. Practical Implications in Statistical Process Control

Time series of counts arise naturally in several applications, namely the manufacturing industry, health care, service industry, insurance, and network analysis. Using control charts for monitoring the underlying count processes is essential to swiftly detect changes in such processes and start preventive or corrective actions (see Weiß [13]). For an overview of control charts for count processes, we refer the reader to Weiß [14].

As noted by Ristić et al. [6], counts with geometric marginal distributions play a *major role* in several areas, for instance reliability, medicine, and precipitation modeling. These counts may refer to the number of *machines waiting for maintenance*, *congenital malformations*, or *thunderstorms in a day*.

In statistical process control, the GINAR(1) process can be used to model, for example, the cumulative counts of conforming items between two nonconforming items when these successive counts are no longer independent, say because the observations are generated by automated high-frequency sampling.

The literature review reveals that no charts have been proposed for monitoring GINAR(1) or GGINAR(1) counts. However, Li et al. [15] proposed a combined jumps chart, a cumulative sum (CUSUM) chart, and a combined exponentially weighted moving average (EWMA) chart for monitoring the NGINAR(1) counts. Furthermore, Li et al. [16] described upper and lower one-sided CUSUM charts for monitoring the mean of ZMGINAR(1) counts.

Let us consider that the following quality control chart is being used to detect decreases in the parameter *p* of the GINAR(1) process.

**Definition** **3.** 
*Let $\left\{X_{t}:t\in N_{0}\right\}$ be a GINAR(1) process. The upper one-sided geometric chart makes use of the set of control statistics $\left\{X_{t}:t\in N\right\}$ and triggers a signal at time t(t∈N) if $X_{t}>U$, where U is a fixed upper control limit (UCL) in $N_{0}$.*


We should bear in mind that the control statistic $X_{t}$ becomes stochastically smaller in the usual sense as *p* increases (see Lemma A4). Consequently and as suggested by (Xie et al. [17] p. 42), it is clear that when an observed value of $X_{t}$ exceeds the UCL of the chart, this should be taken as a sign that the *p* has decreased, that is, an indication of a potential increase in the process mean (1−p)/p.

The performance of the upper one-sided geometric chart is about to be assessed in terms of the run length (RL), the random number of samples collected before a signal is triggered by this control chart. Consequently, the following first passage time of the stochastic process $\left\{X_{t}:t\in N_{0}\right\}$, under the condition that $X_{0}=u\in \left\{0,1,\cdots ,U\right\}$, is a vital performance measure of this chart for monitoring a GINAR(1) process:(8)$$RL^{u}\equiv RL^{u}\left(U\right)=\min\left\{t\in N:X_{t}>U|X_{0}=u\right\},$$
where *u* is a fixed initial value in the set {0,1,⋯,U}.

*U* is chosen in such a way that false alarms are rather infrequent and increases in the process mean (1−p)/p (i.e., decreases in *p*) are detected as quickly as possible. Hence, we should be dealing with a large in-control RL and smaller out-of-control run lengths.

### 3.1. Significance of $P\in {TP}_{2}$

By invoking the first part of Theorem 3.1 of Assaf et al. [18], we can state that the ${TP}_{2}$ character of the TPM of the GINAR(1) process leads to the following result.

**Corollary** **1.** 
*Let $\left\{X_{t}:t\in N_{0}\right\}$ be a GINAR(1) process. Then*

(9)
$$RL^{0}=\min\left\{t\in N:X_{t}>U|X_{0}=0\right\}\in {PF}_{2},$$

*i.e., ${\left[P_{RL^{0}}\left(x+1\right)\right]}^{2}\geq P_{RL^{0}}\left(x\right)\times P_{RL^{0}}\left(x+2\right)$, for $x\in N_{0}$.*


Corollary 1 implies that $RL^{0}$ has an increasing hazard rate $\left(RL^{0}\in IHR\right)$, that is, $\lambda_{RL^{0}}\left(m\right)=P\left(RL^{0}=m\right)/P\left(RL^{0}\geq m\right)$ is a nondecreasing function of m∈N (see Kijima [19] p. 118, Theorem 3.7(ii)). $RL^{0}\in IHR$ means that signaling, given that no observation has previously exceeded the UCL, becomes more likely as we proceed with the collection of observations provided that $X_{0}=0$.

Note, however, that $RL^{u}$ may not be IHR, for u∈{1,⋯,U}. In fact, the second part of Theorem 3.1 of Assaf et al. [18] allows us to state that the p.f. $P_{RL^{U}}\left(l+n\right)$ is ${TP}_{2}$ in *l* and *n*$\left(l,n\in N_{0}\right)$, i.e., $P_{RL^{U}}\left(l+n\right)\times P_{RL^{U}}\left(l^{\prime }+n^{\prime }\right)\geq P_{RL^{U}}\left(l^{\prime }+n\right)\times P_{RL^{U}}\left(l+n^{\prime }\right)$, for $l,n\in N_{0}$$\left(l<l^{\prime },n<n^{\prime }\right)$. As a consequence, ${\left[P_{RL^{U}}\left(x+1\right)\right]}^{2}\leq P_{RL^{U}}\left(x\right)\times P_{RL^{U}}\left(x+2\right)$, for $x\in N_{0}$, thus we can add that $RL^{U}$ has an decreasing hazard rate $\left(RL^{U}\in DHR\right)$.

The next corollary translates the stochastic influence of an increase in the initial value *u* and can be shown to be valid by capitalizing on (Karlin [20] pp. 42–43, Theorem 2.1).

**Corollary** **2.** 
*Let $\left\{X_{t}:t\in N_{0}\right\}$ be a GINAR(1) process. Then, for $u,u^{\prime }\in \left\{0,1,\cdots ,U\right\}$,*

(10)
$$RL^{u^{\prime }}\leq_{lr}RL^{u},\;u\leq u^{\prime }.$$



Let us denote the upper one-sided geometric chart with $X_{0}=u^{\prime }$ (resp. $X_{0}=u$) by Scheme 1 (resp. Scheme 2). Then (Equation 10) can be interpreted as follows: the odds of Scheme 1 signaling at sample *m* against Scheme 2 triggering a signal at the same sample decreases as *m* increases (see [21] p. 5).

Result (Equation 10) seems *quite evident*; nevertheless, it would not be valid if the GINAR(1) process was not governed by a ${TP}_{2}$ TPM.

### 3.2. Other Comparisons of Run Lengths

The stochastic inequality $P\left(p^{\prime },\rho \right)\leq_{st}P\left(p,\rho \right)$, for $0\leq \sqrt{\rho }/\left(\sqrt{\rho }+1\right)<p\leq p^{\prime }<1$, allows us to stochastically compare two GINAR(1) processes. As a matter of fact, by invoking Lemma A4 and Theorem 6.B.32 of (Shaked and Shanthikumar [9] p. 282), we can state the next result.

**Corollary** **3.** 
*Let $\left\{X_{t}\left(p^{\prime },\rho \right):t\in N_{0}\right\}$ and $\left\{X_{t}\left(p,\rho \right):t\in N_{0}\right\}$ two GINAR(1) processes. If $0\leq \sqrt{\rho }/\left(\sqrt{\rho }+1\right)<p\leq p^{\prime }<1$ and the initial states are deterministic $X_{0}=u^{\prime }\leq X_{0}\left(p^{\prime }\right)=u$ or random, say $X_{0}\left(p^{\prime },\rho \right)∼geometric\left(p^{\prime }\right)\leq_{st}X_{0}\left(p,\rho \right)∼geometric\left(p\right)$, then*

(11)
$$\left\{X_{t}\left(p^{\prime },\rho \right):t\in N_{0}\right\}\leq_{st}\left\{X_{t}\left(p,\rho \right):t\in N_{0}\right\}.$$



From (Equation 11) we can infer from (Equation 11) that $X_{1}\left(p^{\prime },\rho \right)\leq_{st}X_{1}\left(p,\rho \right)$.

The next lemma plays a vital role in the comparison of run lengths and is taken from (Shaked and Shanthikumar [9] p. 283).

**Lemma** **1.** 
*If two stochastic processes $\left\{X_{t}:t\in T\right\}$ and $\left\{Y_{t}:t\in T\right\}$ satisfy $\left\{X_{t}:t\in T\right\}\leq_{st}\left\{Y_{t}:t\in T\right\}$ then*

(12)
$$\inf\left\{t\in T:Y_{t}>U\right\}\leq_{st}\inf\left\{t\in T:X_{t}>U\right\}.$$



Lemma 1 states what could be considered obvious: if we are dealing with two ordered stochastic processes in the usual sense, the larger stochastic process in the usual sense exceeds the critical level *U* stochastically sooner also in the usual sense.

By combining Corollary 3 and Lemma 1, we can provide a stochastic flavor to the influence of an increase in *p* not only on $RL^{u}$ but also on another important RL:(13)$$RL^{X_{1}}=\min\left\{t\in N:X_{t}>U|X_{1}\right\},$$
which we coin as *overall run length*, following (Weiß [22] Section 20.2.2). $RL^{X_{1}}$ refers to a first passage time of the stochastic process $\left\{X_{t}:t\in N\right\}$ under the condition that the initial state coincides with the r.v. $X_{1}$. In point of fact, it is reasonable to resort to this performance measure because in practice we do not know $X_{0}$, hence it is plausible to rely, for example, on $X_{1}∼geometric\left(p\right)$.

**Corollary** **4.** 
*The following stochastic ordering results hold for the run lengths of the upper one-sided geometric chart for monitoring GINAR(1) processes:*

(14)
$$\begin{array}{ccc} RL^{u}\left(p,\rho \right) & \leq_{st} & RL^{u^{\prime }}\left(p^{\prime },\rho \right) \end{array}$$


(15)
$$\begin{array}{ccc} RL^{X_{1}\left(p,\rho \right)}\left(p,\rho \right) & \leq_{st} & RL^{X_{1}\left(p^{\prime },\rho \right)}\left(p^{\prime },\rho \right), \end{array}$$

*for $u^{\prime }\leq u$ and $0\leq \sqrt{\rho }/\left(\sqrt{\rho }+1\right)<p\leq p^{\prime }<1$.*


Note that we could have also invoked (Equation 14) and the closure of the usual stochastic order $\leq_{st}$ under mixtures (see Shaked and Shanthikumar [9] p. 6, Theorem 1.A.3.(d)) to prove (Equation 15).

Results (Equation 14) and (Equation 15) mean that the upper one-sided geometric chart for the GINAR(1) process stochastically increases its detection speed (in the usual sense) as the downward shift in *p* becomes more extreme. This stochastic ordering result parallels with the notion of a sequentially repeated uniformly powerful test.

### 3.3. An Illustration

Ristić et al. [6] found that an NGINAR(1) model with estimated parameters ${\hat{p}}_{0}=1/\left(1+0.5872\right)=0.63$ and ${\hat{\rho }}_{0}=0.1650$ adequately described the monthly counts of sex offenses reported in the 21st police car beat in Pittsburgh. This data set comprises 144 observations, starting in January 1990 and ending in December 2001.

Note that the GINAR(1) and NGINAR(1) processes share the same geometric marginal distribution; and, as far as the offense data set is concerned, the value of the Akaike information criterion (AIC) for the NGINAR(1) and GINAR(1) models are very close, namely 302.67 and 303.74, respectively, as (Ristić et al. [6] Table 2) attest. Hence, we are going to consider the upper one-sided geometric chart from Definition 3 with $p_{0}=0.63$ and $\rho_{0}=0.1650$ for monitoring such counts.

An UCL equal to U=5 and an initial state u=0 (resp. u=U) yield an in-control ARL of $E[RL^{0}\left(p_{0},\rho_{0}\right)]≃393.7$ (resp. $E[RL^{U}\left(p_{0},\rho_{0}\right)]≃391.4$). These and other RL-related performance measures used in this subsection are described in Section A.2.

The plots of the hazard rate function in Figure 1 give additional insights into the RL performance of the geometric chart as we proceed with the sampling and to the impact of the adoption of a head start. Indeed, it illustrates two results that follow from Corollary 1: $RL^{0}\left(p_{0},\rho_{0}\right)\in IHR$ and $RL^{U}\left(p_{0},\rho_{0}\right)\in DHR$. This last result suggests that the false-alarm rate conveniently decreases in the first samples when we adopt a head start (u=U>0).

According to Brook and Evans [23], the limiting form of the p.f. of the RL is geometric-like with parameter 1−ξ(p,ρ), where ξ(p,ρ) is the maximum real eigenvalue of $Q\left(p,\rho \right)={\left[p_{ij}\left(p,\rho \right)\right]}_{i,j\in \{0,1,\cdots ,U\}}$, regardless of the initial value *u* of the control statistic $X_{t}$. Therefore, it comes as no surprise that the values of the hazard rate functions of $RL^{0}\left(p_{0},\rho_{0}\right)$ and $RL^{U}\left(p_{0},\rho_{0}\right)$ converge to
(16)$$\lim_{m\to +\infty }\lambda_{RL^{0}\left(p_{0},\rho_{0}\right)}\left(m\right)=\lim_{m\to +\infty }\lambda_{RL^{U}\left(p_{0},\rho_{0}\right)}\left(m\right)=1-\xi \left(p,\rho \right)≃0.002541,$$
as suggested by Figure 1.

Furthermore, the hazard rate function of $RL^{0}\left(p_{0},\rho_{0}\right)$ is pointwise below the one of $RL^{U}\left(p_{0},\rho_{0}\right)$ because Corollary 2 establishes that $RL^{U}\left(p_{0},\rho_{0}\right)\leq_{lr}RL^{0}\left(p_{0},\rho_{0}\right)$ and this result in turn implies $RL^{U}\left(p_{0},\rho_{0}\right)\leq_{hr}RL^{0}\left(p_{0},\rho_{0}\right)$, that is, $\lambda_{RL^{U}\left(p_{0},\rho_{0}\right)}\left(m\right)\geq \lambda_{RL^{0}\left(p_{0},\rho_{0}\right)}\left(m\right)$, for m∈N (see Definition A4).

We now illustrate the first result of Corollary 4 and also of a consequence of its second result: $RL^{0}\left(p,\rho \right)\leq_{st}RL^{0}\left(p^{\prime },\rho \right)$, for $0\leq \sqrt{\rho }/\left(\sqrt{\rho }+1\right)<p\leq p^{\prime }<1$; $E[RL^{X_{1}\left(p,\rho \right)}\left(p,\rho \right)]$ is an increasing function of *p* in the interval, $\left(\sqrt{\rho }/\left(\sqrt{\rho }+1\right),1\right)$.

In the left panel of Figure 2, we plotted the survival functions of $RL^{0}\left(0.9p_{0},\rho_{0}\right)$ and $RL^{0}\left(p_{0},\rho_{0}\right)$.

Since $RL^{0}\left(0.9p_{0},\rho \right)\leq_{st}RL^{0}\left(p_{0},\rho \right)$, the plot of survival function of $RL^{0}\left(0.9p_{0},\rho \right)$ is pointwise below the one of $RL^{0}\left(p_{0},\rho \right)$, as Figure 2 plainly demonstrates. Hence, the number of samples taken until the detection of a 10% decrease in *p* by the upper one-sided geometric chart is indeed stochastically smaller than the number of samples we collect until this chart emits a false alarm.

The right panel of Figure 2 refers to the overall ARL function, $E[RL^{X_{1}}\left(p,\rho_{0}\right)]$, for $\sqrt{\rho }/\left(\sqrt{\rho }+1\right)<p\leq p_{0}$. It increases with *p* in this particular interval from $E[RL^{X_{1}}\left(\sqrt{\rho_{0}}/\left(\sqrt{\rho_{0}}+1\right),\rho_{0}\right)]≃8.3$ to $E[RL^{X_{1}}\left(p_{0},\rho_{0}\right)]≃393.5$. We ought to note that it increases further when we take $p\in (p_{0},1)$, therefore the upper one-sided geometric chart cannot detect increases in *p* in an expedient manner, as we have anticipated.

We wrote a program for Mathematica 10.3 (Wolfram [24]) to produce all the graphs and results in this subsection.

## 4. Concluding Remarks

As expertly put by Montgomery and Mastrangelo [25], the independence assumption is often violated in practice. As a consequence, we often deal with discrete-valued time series, namely when we are dealing with very high sampling rates, as suggested by Weiß and Testik [26], and Rakitzis et al. [27].

In this paper, we considered the GINAR(1) count process, resorted to stochastic ordering to prove two features of its TPM, and discussed the implications of these two traits on RL-related performance measures of an upper one-sided geometric control chart that accounts for the autocorrelated character of such process.

For example: the ${TP}_{2}$ character of the TPM of the GINAR(1) process implies an IHR behaviour of the run length $RL^{0}$ of that same chart; the run length $RL^{u}$ and the overall run length $RL^{X_{1}}$ stochastically increase in the usual sense in the interval $\left(\sqrt{\rho_{0}}/\left(\sqrt{\rho_{0}}+1\right),1\right)$.

These features of the GINAR(1) process and the associated results are comparable to the ones derived by (Morais [21] Section 3.2) and Morais and Pacheco [8,28].

It is important to note that the notion of stochastically monotone matrices in the usual sense was introduced by Daley [29] for real-valued discrete-time Markov chains. Moreover, Karlin [20] implicitly states that a ${TP}_{2}$ TPM possesses a monotone likelihood ratio property and, thus, virtually defines stochastically monotone Markov chains in the likelihood ratio sense. Furthermore, the comparison of counting processes and queues in the usual sense can be traced back, for instance, to Whitt [30] and the multivariate likelihood ratio order of random vectors (or ${TP}_{2}$ order) is discussed, for example, by (Shaked and Shanthikumar [9] pp. 298–305).

Coincidentally, the stochastic order in the likelihood sense for stochastic processes or TPM has not been defined up to now, as far as we have investigated. For this reason and the fact that the $\leq_{lr}$ order is not closed under mixtures (see Shaked and Shanthikumar [31] p. 33), we did not state or prove the $\leq_{lr}$ analogue of the two results in Corollary 4.

We also failed to prove that $P\left(p,\rho^{\prime }\right)\leq_{st}P\left(p,\rho \right)$, for $0\leq \rho \leq \rho^{\prime }<1$, because of two opposing stochastic behaviors of the summands $\left(\rho \circ X_{t}|X_{t}=i\right)$ and $\varepsilon_{t+1}$: the r.v. binomial(i,ρ) (resp. BG(p,ρ)) stochastically increases (resp. decreases) with ρ in the likelihood ratio sense. Had we proven that result, we could have concluded that the larger the upward shifts in the autocorrelation parameter, the longer it takes the upper one-sided geometric chart to detect such a change in ρ.

It would be pertinent to investigate the stochastic properties of the RL and overall RL of lower one-sided geometric charts for detecting increases in the parameter *p* of a GINAR(1) process.

Another possibility of further work which certainly deserves some consideration is to investigate the extension of Theorems 1 and 2 to the NGINAR(1) process, the novel geometric INAR(1) process proposed by Guerrero et al. [32], or the new INAR(1) process with Poisson binomial-exponential 2 innovations studied by Zhang et al. [33], and assess the impact of these two results in the RL performance of upper one-sided charts for monitoring such autocorrelated geometric counts.

We ought to mention that deriving results similar to (Equation 6) and (Equation 7) seems to be very unlikely for the mixed generalized Poisson INAR process [34]. This follows from the fact that the generalized Poisson distribution has not a ${PF}_{2}$ p.f.

## Figures and Tables

**Figure 1 entropy-25-00444-f001:**
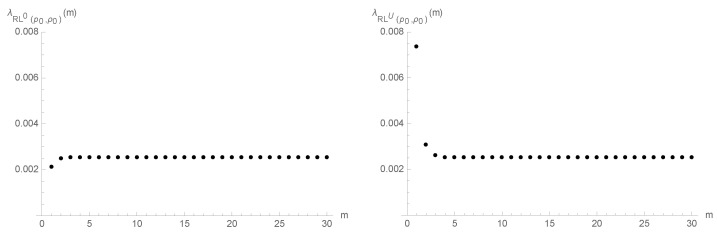
Hazard rate functions of $RL^{0}\left(p_{0},\rho_{0}\right)$ and $RL^{U}\left(p_{0},\rho_{0}\right)$.

**Figure 2 entropy-25-00444-f002:**
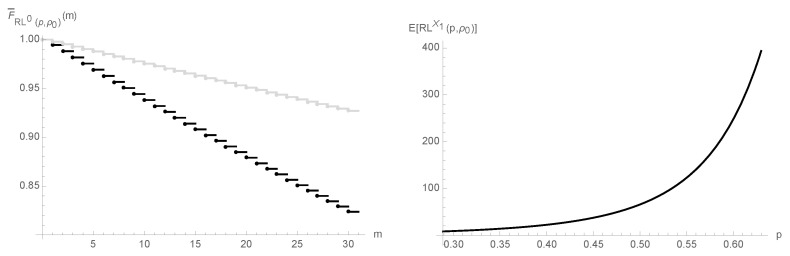
Survival function of $RL^{0}\left(p,\rho_{0}\right)$, for $p=0.9p_{0}$ and $p=p_{0}$ (black and gray solid lines); overall ARL function, $E[RL^{X_{1}}\left(p,\rho_{0}\right)]$, for $\sqrt{\rho }/\left(\sqrt{\rho }+1\right)<p\leq p_{0}$.

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
