# Peer review of "Two Features of the GINAR(1) Process and Their Impact on the Run-Length Performance of Geometric Control Charts"

_entropy, 2023, doi:10.3390/e25030444_

Round 1

Reviewer 1 Report

The paper is interesting, and it is well written and designed.

1. Is the marginal geometric distribution critical? Usually we are familiar with the marginal Poisson INAR(1) model.

2. Guerrero et al. (2022, Stochastic Models, 38, 70-90, https://doi.org/10.1080/15326349.2021.1977141) proposed a new geometric INAR(1) model, can the results here be applied to this paper?

3. The method discussed here relies on the INAR(1) model with a specific marginal distribution, but we know that more INAR(1) models are defined without giving marginal distribution, such as Huang et al.(2023) and Zhang et al. (2023), what we can do in this situation? A brief discussion is needed.

Huang, J., Zhu, F. and Deng, D. (2023). A mixed generalized Poisson INAR model with applications. Journal of Statistical Computation and Simulation, forthcoming.

Zhang, J., Zhu, F. and Mamode Khan, N. (2023). A new INAR model based on Poisson-BE2 innovations. Communications in Statistics-Theory and Methods, forthcoming.

Reviewer 2 Report

The article discusses the potential use of the geometric first-order integer-valued autoregressive process (GINAR(1)) in modelling relevant discrete-valued time series, especially in statistical process control. The authors utilise stochastic ordering to demonstrate that the GINAR(1) process is a discrete-time Markov chain.

The authors also employ stochastic ordering to compare transition matrices of pairs of GINAR(1) processes with different p values. The implications of these stochastic ordering results are then assessed and illustrated, particularly regarding the properties of the run length of geometric charts for monitoring GINAR(1) counts.

This article provides a detailed and thorough analysis of the GINAR(1) process and its potential use in statistical process control. The utilisation of stochastic ordering and TP2 transition matrices adds a valuable contribution to understanding the properties of GINAR(1) and its application in real-world scenarios. In addition, the article is well-written and well-organised, making it a valuable reference for those interested in statistical process control of integer-valued time series.

The paper is clearly written from a theoretical perspective. Nevertheless, I think that possible use cases could be discussed. For which application areas is the monitoring of integer-valued processes interesting? Are there any typical applications where GINAR(1) models fit well? I am also thinking about whether "modern" issues, such as monitoring bike-sharing data could be interesting. For bike-sharing stations, you often observe counts which are small (sometimes ranging from 5-15 bikes in the rush hours) and which are correlated across time.

The subdivision of the paragraphs could undoubtedly be improved in some places; for example, some paragraphs consist of only one sentence. Also, I noticed that in some places, a paragraph starts directly on the following line (with indentation) and sometimes has a space. Is it used differently? Other sentences begin with a reference to a formula (line 135, "(10) seem quite evident"). I would suggest always starting sentences with a word for readability. Finally, I would recommend not interrupting the roadmap of the paper at the end of the introduction with the two theorems. 

P.S.: I have tried to follow the style of reviews of mdpi journals; meaning, to provide rather general comments.

Reviewer 3 Report

Dear Authors, 

thank you for the possibility to report on your paper entitled: "Two features of the GINAR(1) process and their impact on the run length performance of geometric control charts". The paper is interesting, and nicely written. It provides two theorems concerning orderings of two integer-valued processes. Furthermore it build up the control chart for detecting the changes in $\rho$.

Taking into account the general scope of the Journal, authors need to specify more concepts used in the paper. For example, I wonder if readers of the paper are familiar with the abbreviations like DTMC and TPM. It would be also good if the INAR process is also defined (and not directly GINAR). I would also welcome a smoother introduction, using potential applications and a literature overview. It is unknown, whether there are other papers investigating control charts of the INAR or GINAR.

I would like to see the support of the results based on some simulations. 

Please also give a feeling of the data used in the empirical study. How large are the values used for the study? This question helps me to answer, whether an integer-based model is needed and whether an approximation by the real-valued one cannot be used. 

Small typo is on line 82, where after expectation we have an opening round bracken, but the closing rectangular one. Same typo is on line 57.

Yours, 

Referee

Round 2

Reviewer 1 Report

I am satisfied with the revision.

Author Response

Dear Reviewer,
Once more, thank you very much for your valuable comments on our manuscript and your assistance.
Cordially,
Manuel Cabral Morais

Reviewer 2 Report

I have no further comments. Congratulations on your excellent contribution to the special issue.

Author Response

(The authors gave the same response as above.)

Reviewer 3 Report

Dear Authors, 

thank you for adressing my points. I have no further questions. 

Yours, 

Referee

Author Response

(The authors gave the same response as above.)
